Adaptive quality of service for packet loss reduction using OpenFlow meters

Deo Krishneel
http://orcid.org/0000-0002-2378-7745 Chaudhary Kaylash kaylash.chaudhary@usp.ac.fj
Assaf Mansour
School of Information Technology, Engineering, Mathematics and Physics, The University of the South Pacific , Suva, Rewa , Fiji
Lee Ahyoung
Electronic publication date: 2024 Apr 4
Publication date: 2024
Volume: 10
Electronic Location ID: e1848
Received 2023 Jul 18; Accepted 2024 Jan 10
Copyright: © 2024 Deo et al.
Copyright year: 2024
Copyright holder: Deo et al.
License: This is an open access article distributed under the terms of the Creative Commons Attribution License, which permits unrestricted use, distribution, reproduction and adaptation in any medium and for any purpose provided that it is properly attributed. For attribution, the original author(s), title, publication source (PeerJ Computer Science) and either DOI or URL of the article must be cited.
License URL: https://creativecommons.org/licenses/by/4.0/

Keywords: SDN, Loss reduction, Adaptive QoS, OpenFlow meter

Funding: The authors received no funding for this work.

==============================
Quality of Service (QoS) is a mechanism used in computer networks to prioritize, classify, and treat packets differently based on certain criteria. This helps the switching devices to schedule and reorder packets if there is congestion in the network. Edge routers experience high traffic congestion as a result of traffic aggregation from the internal network devices. A router can have multiple QoS classes configured, and each class could experience traffic at various rates. However, when a QoS class is underperforming or needs more bandwidth, some bandwidth can be borrowed or leased out to another QoS class to ensure the link is utilized to maximum capacity and the highest throughput is achieved. This article proposes a bandwidth allocation and distribution algorithm that purely uses the flow statistics from the OpenFlow switches to allocate bandwidth to different QoS classes optimally based on their current requirement. The algorithm does not guarantee in advance that the packet loss will be minimized but does guarantee the initial minimum bandwidth allocation. It adjusts the flows’ rates with the aim to increase their current throughput. The algorithm uses the Software Defined Networking (SDN) controller’s flow monitoring component to query the flow statistics from the switch to first approximate the traffic flow rate and then calculate the optimal bandwidth values to assign to each QoS class. The proposed algorithms will be applied to certain switches in the path with the assumption that all the switches are OpenFlow compatible. The algorithm’s performance was compared with the Adaptive Quality of Service (AQoS) algorithm over various traffic scenarios. The results show that the proposed algorithm achieves an average of 9% performance gain compared to the AQoS algorithm.

Introduction

Packet-switched networks first came into existence with the development of ARPANET to allow remote access to computers. The development of ARPANET commenced in the 1960s to meet the communication requirements without the need of telephone lines (Roberts, 1986). Since its development, the internet has been an elite amongst the military, universities, and the research community. By the 1990s, the internet began to percolate in the society as a luxurious item, following the continuous research and development from the research community (Naughton, 2016). Many internet-enabled technologies started emerging by the 2000s, such as Web 2.0, mobile connectivity, surveillance, and social media, to name a few. As more internet-enabled technologies came into existence, quality of service (QoS) played a vital role in everyday networks as the need for more bandwidth became a problem.

Quality of service was first defined in 1994 by the International Telecommunication Union (ITU) (International Telecommunication Union (ITU), 2008). It is a mechanism that can provide differential treatment to different flows based on their priority. In general, every traffic passing through the network device is treated on a best effort basis, i.e., all the traffic flows are treated equally and have an equal chance of being dropped should congestion occur at the forwarding device. However, some traffic, such as voice over IP (VoIP) and multiplayer online gaming, are sensitive to delay (Yildirim & Radcliffe, 2010) and require a fixed flow rate. Traditionally, QoS was implemented using two major approaches; integrated services (IntServ) (Braden, Shenkar & Clark, 1994) and differentiated services (DiffServ) (Nichols, Jacobson & Zhang, 1999). IntServ provides fine-grained flow-based control over the system and requires that every router in the path from source to destination implement IntServ. On the other hand, DiffServ is a coarse-grained class-based control system that works on classifying and marking packets to specific classes. Each router in the path is configured to mark and classify traffic to a class. The router can then treat each traffic class differently based on its priority.

With the advent of software defined networking (SDN) (Kreutz et al., 2015) and the OpenFlow (Open Networking Foundation, 2012) protocol, managing QoS became easier (Boucadair & Jacquenet, 2014) as it introduced new methods of configuring and managing QoS rules. SDN is an approach to network management and programmability via a central network controller (Malik et al., 2019). It separates the control and data plane of the forwarding devices and manages all the forwarding decisions centrally on the controller. The controller can query and update information on the devices and has a global view of the network. SDN introduced the concept of meters in OpenFlow 1.3 (Open Networking Foundation, 2012) that allows the controller to rate limit flows based on certain match fields and priority. A meter measures and controls the rate of traffic flowing through the device using a meter table. A meter table contains entries that dictate how the flows are matched to a meter on a per-flow basis. The meter table can be used to calculate the rate of flow using the counters in the meter table, and different actions can be applied to flows when the rate exceeds the set limit.

OpenFlow meter allows flexible bandwidth adjustment and can easily adapt to changing bandwidth requirements. One of the major problems addressed by this feature of OpenFlow meters is bandwidth allocation and distribution (Boley, Jung & Kettimuthu, 2017). Traditional methods of QoS configuration do not allow a flexible way of bandwidth allocation resulting in the link not being utilized to full capacity. It also does not allow other QoS flows to expand into the unused bandwidth of the other flows, causing packet losses for flows that need more bandwidth while the link can accommodate these extra requirements. Gateways and edge devices experience high traffic congestion due to traffic aggregation from the lower tier network devices. The devices can have multiple QoS classes configured, and each class could experience traffic at various rates. When a QoS class is using less than its configured bandwidth or needs more bandwidth, some bandwidth can be borrowed or leased out to another QoS class to ensure maximum link utiliation and throughput. Several works (Boley, Jung & Kettimuthu, 2017; Hong, Kandula & Mahajan, 2013; Jain, Kumar & Mandal, 2013) have attempted to address this issue using OpenFlow meters and hybrid approaches. Therefore, this article also attempts to approach the same problem. However, it looks at the traffic at a switch that experiences the highest congestion level and when it is bursty. It attempts to minimize the packet loss on a per-flow or a per-QoS class basis while also ensuring that high-priority requirements are satisfied by the unused bandwidth of other flows. The article presents an algorithm that implements these constraints and tests the algorithm with different flow rates per class to test the effectiveness of the algorithm.

The authors of this article propose, implement, and test the bandwidth allocation and redistribution algorithm for bursty flows at the highest congestion switch, such as the edge router. It takes advantage of the controller’s ability to query and monitor flow statistics. The algorithm calculates the excess bandwidth required by each QoS class and tries to find an empty QoS class that can satisfy the growing need. If multiple classes need more bandwidth, the algorithm carefully chooses the amount of bandwidth to allocate to each class while keeping a little margin for it grow to avoid packet losses. One of the issues with a resource shared network is fairness (Hwang & Chi, 2003). The algorithm employs a combination of max-min and weighted fairness mechanism to implement fairness in the proposed algorithm. The algorithm also ensures that the QoS class whose bandwidth had been previously borrowed are returned when the classes need more bandwidth, ensuring that the minimum guaranteed bandwidth is given to each QoS class.

The contributions in this article can be summarized as follows: Framework: The authors propose a framework that can easily integrate into Ryu controller as a controller application. The framework monitors the flows in real-time and strategically uses the statistics collected to allocate, distribute, and claim that bandwidth at the highest congestion switch. In literature, several works have attempted to make applications that focus on maximizing the overall throughput in different contexts. However, according to the authors' knowledge, no study has focused on maximizing the per-flow throughput while optimally adjusting the per-flow or per-class QoS in bursty network conditions.

Design and implement three components: a traffic monitoring component, a rate approximation component, and a bandwidth allocation and distribution component. Other works (Boley, Jung & Kettimuthu, 2017) have proposed a similar component; however, this article focuses on optimizing the logic of the components to achieve the article’s objectives.

Evaluation: The algorithm is evaluated through extensive and multiple simulations using virtual networks in Mininet. The analysis shows that the proposed algorithm achieves higher throughput and incurs less packet loss per-flow or per-QoS class.

The rest of the article is organized as follows: “Related Work” presents the related work on similar applications built using OpenFlow meters to configure and manage QoS or allocate bandwidth. “Problem Statement and Objectives” presents the problem statement as well the objectives. The bandwidth allocation algorithms overview, design, and implementation are discussed in “Bandwidth Allocation and Redistribution Methods”. “Testbed Setup” presents the testbed setup on which the entire experiment was setup. In “Methodology”, we discuss the methodology that was used to gather, analyze and test our algorithms. The algorithm is evaluated in “Results and Discussion”, showing the effectiveness of the algorithm, its loss per class and bandwidth distribution. Finally, the article concludes in “Conclusion”, discussing the contributions and recommendations for future work.

Related work

SDN has addressed many network programmability problems (Hu et al., 2013; Zhou, Zhu & Xiao, 2015; Long et al., 2013; Chou et al., 2014; Yang & Yeung, 2020; Ze & Yeung, 2017; Kaur, Kaur & Gupta, 2016; Kiadehi, Rahmani & Molahosseini, 2021) and has provided a flexible way of managing networks. Some of these include load balancing applications such as the work in Hwang & Chi (2003), which implements load balancing in wide-area OpenFlow networks, controller load balancing (Hu et al., 2013) to reduce overhead message exchange traffic, path switching to dynamically balance the traffic during data transmission in data centers (Zhou, Zhu & Xiao, 2015), distributing large dataset to clients using genetic algorithm (Long et al., 2013), and load balancers which converts the data plane into load balancers by employing certain strategies in the POX controller (Ze & Yeung, 2017). Flow monitoring was another component that reaped the benefits of SDN. The work in Yang & Yeung (2020) proposed a flow monitoring scheme that allows an SDN controller to periodically collect statistics of flows in the network. Another solution presented a method of minimizing network bandwidth (Chou et al., 2014) for flow monitoring by polling some statistics to prevent overwhelming the OpenFlow switches. SDN has not only leveraged the possibilities in organizational networks but also in IoT by providing fault tolerant solutions (Kiadehi, Rahmani & Molahosseini, 2021).

Along with the features provided by traditional networks, SDN provides a feature rich Controller-Data Plane Interface (C-DPI) that can tune the forwarding rules at a more granular level. QoS is one of the domains that have become feature rich with the introduction of SDN. QoS has benefited in the following areas: multimedia flows routing mechanisms, inter-domain routing mechanisms, resource reservation mechanisms, queue management and scheduling mechanisms, Quality of Experience (QoE)-aware mechanisms, network monitoring mechanisms, and other QoS-centric mechanisms such as virtualization-based QoS provisioning and QoS policy management (Karakus & Durresi, 2017). This section will first outline some of these works carried out in QoS domain using SDN, then look at some work related to this article and present a brief comparison with the proposed algorithm.

The work in Akella & Xiong (2014) proposes an approach to ensure end-to-end QoS guarantees to multiple users on the cloud platform. Cloud computing allows users to access and share remote files without owning physical devices. The work presents an expression that selects a path to carry the traffic from the cloud provider to the user. In Tomovic, Prasad & Radusinovic (2014), the authors present an SDN framework that programs the network devices to provide the relevant QoS requirements for multimedia applications. It provides bandwidth guarantees for priority flows and provides a mechanism to monitor the network state and resource levels. The authors in Amiri, Osman & Shirmohammadi (2018) propose a bandwidth allocation mechanism that intends to prioritize gaming flows with the data center to ensure that the delay sensitive game applications are not affected. It uses machine learning to classify different traffic to improve the QoS needs of the data center network. The work in Breiki, Zhou & Luo (2020) introduces a solution to enhance the QoS performance for streaming mission-critical video data in an OpenFlow network. The solution, called the Meter Band Rate Evaluator (MBE), uses a description language to optimize the QoS capabilities in the OpenFlow network. While these works provide insight into specific applications, some solutions look at the traffic as a whole for providing QoS solutions. Table 1 summarizes these work and compares their similarities and differences with the proposed approach.

Table 1 Literature summary (Hong, Kandula & Mahajan, 2013; Jain, Kumar & Mandal, 2013; Amiri, Osman & Shirmohammadi, 2017; Al-Haddad et al., 2021; Sminesh, Kanaga & Ranjitha, 2017; Torres et al., 2020; Elahi et al., 2020; Qadeer, Lee & Tsukamoto, 2021; Keshav et al., 2021).

Paper	Summary	Similarity/difference/demerit	
Hong, Kandula & Mahajan (2013)	This work presents a method to increase the link utilization level between data center networks. Traffic going from one data center to the other passes through multiple switches on the path. It controls the amount of traffic, its type and the frequency at which it is sent.	Although a similar approach, the proposed algorithm focuses on edge devices to share the available link bandwidth.	
Amiri, Osman & Shirmohammadi (2017)	The authors present a network bandwidth allocation scheme that attempts to minimize the delay as well as maximize the throughput. The work is targeted at cloud gaming where the users are located remotely on thin clients and access the game resources via the network. It provides some fairness to each user to ensure that the right amount of bandwidth is allocated to each game user.	The work considers the delay in the parameter as well as it a game aware system however, the proposed algorithm considers the packet loss only to maximize the throughput.	
Al-Haddad et al. (2021)	The authors propose two algorithms for optimizing and solving networking congestion issues. It implements the Weighted Fair Queuing (WFQ) mechanism in SDN to attempt a reduction in network congestion. Traffic is first made to follow a set rate using WFQ to make queues for the packets. The proposed method also uses a buffer to decide whether to put the packets into the queue for smooth traffic flow.	The work in Al-Haddad et al. (2021) uses WFQ technique and also employs buffer management. The proposed algorithm only uses meter bands to shape the traffic.	
Sminesh, Kanaga & Ranjitha (2017)	The researchers propose a proactive method of reducing packet loss and congestion by routing the traffic through a different path if the current link experiences congestion. The controller monitors the traffic on each link and determines whether the link is likely to experience congestion. Based on the controller's discretion, a new route is planned, and traffic is routed via the new link.	The proposed algorithm is deployed at a point in the network with only one exit path. The work in Sminesh, Kanaga & Ranjitha (2017) reduces the congestion by rerouting the traffic through another path.	
Jain, Kumar & Mandal (2013)	The work in presents a design and implementation of a private WAN that connects all of Google’s data centers. It addresses the bandwidth management problem by allocating bandwidth to different applications based on their priority with the aim of maximizing the average bandwidth. These solutions provide great insight into the problems they solve.	In contrast, this article looks at how the bandwidth can be allocated to each flow at a device and claimed back when the need for more bandwidth arises while also ensuring there’s least packet loss for each QoS class.	
Torres et al. (2020)	The most recent works have focused not only on the current domain but also in 5G and IoT. The work in Torres et al. (2020) addresses the challenges posed by complex and diverse communication networks environments such as 5G and cloud. It introduces a framework called BAMSDN (Bandwidth Allocation Model through Software-Defined Networking) which focuses on dynamically allocating bandwidth resources using OpenFlow.	The work here is applied in the context of high speed IoT network. The proposed algorithm was not targeted to IoT systems however it can easily be adapted to IoT networks.	
Elahi et al. (2020)	The authors present a bandwidth allocation mechanism that maximizes the network link utilization by user-defined fair allocation of spare bandwidth. It guarantees fairness using the min-max scheme.	This work here is very similar to the proposed algorithm however, the proposed algorithm, along with ensuring minimum bandwidth and sharing the spare band, leaves some margin for the traffic to grow in case of sudden demands.	
Qadeer, Lee & Tsukamoto (2021)	The authors propose a similar approach for bandwidth allocation in IoT. The approach is based on Heuristic Reinforcement Learning (HRL) for flow-level dynamic bandwidth allocation using OpenFlow meters to improve network performance and to validate an end-to-end scenario.	The authors in this work use AI techniques on an end-to-end path whereas the proposed work is applied on a single device.	
Keshav et al. (2021)	Another work on 5G proposed a method of dynamically prioritizing bandwidth to handle multimedia applications with varying requirements. It uses mobile agents to gather the required resource requirements and periodically use this to allocate bandwidth to these applications.	This work uses dependent agents to function whereas the proposed algorithm is a standalone system that uses the data from the OpenFlow switch.	

These solutions provide great insight into the problems they solve. In contrast, this article looks at how the bandwidth can be allocated to each flow at a device and claimed back when the need for more bandwidth arises while also ensuring there’s least packet loss for each QoS class.

In summary, OpenFlow has been introduced to various applications including load balancers for networks as well as SDN controllers. Flow monitoring, bandwidth optimization, end-to-end QoS services, and network device programming to meet various QoS needs have been one of the major focus for OpenFlow. Consumer applications have also benefited from OpenFlow. This includes online gaming and multimedia streaming. While these have been some of the end products of OpenFlow, there have been a lot of research on the underlying complexity that powers the end applications. Various studies have explored the flow statistics to improve QoS with different objectives. The work most similar to this article is AQoS presented in Boley, Jung & Kettimuthu (2017). The authors of AQoS present an algorithm that allows other flows to expand and use the bandwidth that are not currently being used by other high-priority traffic while also ensuring that the initially assigned bandwidth is available for use when needed by those high-priority flows. If there are flows that are not using their allocated bandwidth, the remaining bandwidth is allocated to some other flow that can use it. The borrowed bandwidth is returned to the original flow if the need for more bandwidth arises. This article also allows flows to expand and use other flows’ bandwidth allocation; however, it differs in two ways: it does not fully exhaust the underperforming QoS flows and leaves some room for the borrowed flows to grow incase urgent need for more bandwidth arises. The work proposed in Boley, Jung & Kettimuthu (2017) aims to achieve a similar objective to this article and has proven to perform better than the existing QoS configuration methods (IntServ and DiffServ). Therefore, the proposed algorithms will be compared for performance and throughputs with AQoS.

Problem statement and objectives

The proposed algorithm can be applied to a single switch and an end-to-end path in a network from source to destination; however, it only targets the device with the highest traffic congestion, such as the edge router. The edge router in the network is loaded with the task to meet the service level agreement (SLA) requirements with the service provider and might need to drop packets, or rate limit should congestion occur at the egress port.

It is assumed that the switches are OpenFlow compatible and connected to an OpenFlow controller. Traffic flowing through the switch is first categorized into different classes of traffic based on their priority and QoS levels using the flow table level matching. To simulate the traffic at the switch, every flow entering the switch is categorized into N classes of traffic, with each class sharing a portion of the total egress link bandwidth. Each of the N classes gets A1,A2,…ANkbps of pre-assigned bandwidth, respectively.

The authors address the problem of bandwidth allocation, redistribution and sharing by allowing each QoS flow to expand into other flows when they are not in use while also incurring least packet loss. Other studies have done similar work; however, this article looks at the same problem when the traffic is bursty and minimises the packet loss. In this case, traffic rate approximation and allocation can become challenging.

The algorithm can be defined to best handle the distribution and reallocation of bandwidth if it meets the following conditions:

The allocated rate remains the same when the individual classes’ ingress rate is below the configured rate as well as the cumulative rate is less than the total allocated rates. This obeys and ensures that the min-max fairness is implemented. The algorithm ensures that the minimum set bandwidth is given to each flow as initially defined by the SLA. Mathematically,

(1) Rqi=AqiwhenCqi<Aqi∧∑qi=1qNCqi<∑qi=1qN⁡Aqi,∀qi∈{q1,...,qN}

where; Rqi is the new rate of the ith QoS, Aqi is the pre-configured rate of the ith QoS, and Cqi is the current rate of the ith QoS,

The allocation resets to the pre-configured rate when the individual classes ingress rate exceeds the configured rate as well as the cumulative rate is more than the total allocated rates. This again conforms to the min-max fairness policy. If all the flows are exceeding their set rates, and packet loss is inevitable due to the link being overutilized, the algorithm resets their rates to the initial SLA to guarantee that the minimum required bandwidth is given to each flow. Mathematically,

(2) Rqi=AqiwhenCqi>Aqi∧∑qi=1qNCqi>∑qi=1qN⁡Aqi,∀qi∈{q1,...,qN}

where; Rqi is the new rate of the ith QoS, Aqi is the pre-configured rate of the ith QoS, and Cqi is the current rate of the ith QoS,

If one or more classes ingress rate is exceeding the configured rate and other class is underperforming, the excess bandwidth requirement is satisfied by the underperforming classes. This employs the weighted fairness mechanism to allocate flows to those in need. The excess traffic is divided equally onto the free space or divided based on the priority of the classes. Mathematically,

(3) Rqi= Aqi+ x.Aqj, i ≠j, x∈ (0, 1) when Cqi> Aqi,           Cqj< Aqj ∃< qNqi,  ∀qi,qj∈   {q1,..., qN}

where; Rqi is the new rate of the ith QoS, Aqi is the pre-configured rate of the ith QoS, Cqi is the current rate of the ith QoS, and x is the priority or chosen value of the fraction of other QoS’ bandwidth allocation.

The packet loss incurred by each classes’ meter has to be minimized.

(4) loss=min(Lqi),∀qi∈{q1,...,qN}

where; Lqi is the packet loss incurred by the ith QoS.

Therefore, the main objectives of the article are: Design and implement an algorithm that meets the above conditions to allocate, redistribute, and claim the bandwidth of different QoS classes when a class’s rate exceeds its configured rate and allows it to use other class’ free bandwidth.

Design and implement an algorithm that incurs the least packet loss (maximum throughput) when allocating and redistributing bandwidth.

Apply the algorithm on an OpenFlow network to determine its effectiveness.

Bandwidth allocation and redistribution methods

Algorithm overview

Certain forwarding devices in a network receive more traffic than others and may need to drop or delay the packets due to congestion. With a QoS policy in place, the device can allocate more bandwidth to certain types of flows based on their priority. The bandwidth allocation, redistribution and sharing problem is tackled by first approximating the flow rate. The proposed algorithm then calculates the optimal rate for each QoS class such that the loss per QoS class is minimum and the overall throughput is maximized.

In broad terms, the proposed algorithm addresses the bandwidth allocation problem and reconfigures the switching fabric using the following strategies. Pocket last n immediate flows from the flow graph.

Determine the flow rate of each QoS class or flow.

Rank the QoS classes that need more bandwidth or are underperforming.

Calculate the new rates for each QoS class.

Push the configuration down to the switching fabric.

Algorithm design

Various SDN controllers provide a rich set of APIs to allow communication between the OpenFlow devices and the OpenFlow controller. The proposed algorithm was designed and built using the Ryu SDN controller as a controller application. The allocation application resides along with other applications on the controller, such as the simple_switch, simple_monitor, rest_topology, rest_firewall, etc., and can easily be integrated into the current network infrastructure. The flows under consideration are installed with a different priority to differentiate them from the control packets flowing in the network. The controller queries the switches using the OFPFlowStatsRequest API and analyses the response to calculate the rate and determine the optimal allocation rates. The proposed algorithm is then applied to the individual flow rates to calculate the new rates for each QoS class.

Algorithm implementation

The data received from the switches are first used to calculate an approximate rate at which the traffic is flowing. The current rate of flow is calculated for every flow passing through the flow table or as per the different QoS classes. The switch also has a predetermined rate set that the QoS flows must follow. The allocated and the current calculated rates are fed to Algorithm 1, running as a controller application.

Algorithm 1 meterAllocation(allocated, current).

1  need = unused = {}	
2	
3   for ids in allocated:	
4     if current[ids] >= allocated[ids]:	
5       need[ids] = current[ids] - allocated[ids]	
6     else	
7       need[ids] = 0	
8     if current[ids] < allocated[ids]:	
9      unused[ids] = allocated[ids] - current[ids]	
10    else	
11      unused [ids] = 0	
12    end else	
13  end for	
14	
15  if sum(need) <= sum(unused):	
16     new_rate = allocUnder(allocated, current, need, unused)	
17  else	
18     new_rate = allocOver(allocated, current, need, unused)	
19  end else	
20	
21  return new_rate	

Algorithm 1

Algorithm 1 calculates the needed and unused bandwidth for each QoS flow based on their current and allocated amounts (lines 3–13). The requirement of each QoS class can vary depending on the link utilization. The new rate calculations are different when the link is under utilized than when it is over utilized or at maximum capacity. These cases are handled by two sub procedures (Algorithms 2 and 3) to calculate and return the new flow rate based on whether the total unused bandwidth can satisfy the current needed amount by all the QoS flows.

Algorithm 2 allocOver(allocated, current, need, unused).

1   global be	
2   new_rate = p_need = {}	
3   total_need = sum(need) – need[be]	
4	
5   for ids in need:	
6     new_rate[ids] = min(allocated[ids], current[ids])	
7     if total_need == 0: p_need[ids] = 0	
8     else p_need[ids] = need[ids] / total_need	
9   end for	
10	
11   total_unused = rem_unused = sum(unused)	
12   [need, new_rate, rem_unused] = allocation(need, new_rate, be, p_need, total_unused, rem_unused)	
13   total_need = sum(need) – need[be]	
14	
15   if total_need > 0:	
16     be_available = allocated[be] - unused[be]	
17     if total_need >= be_available:	
18       new_rate[be] = 0	
19       for ids in need:	
20         if total_need == 0: p_need[ids] = 0	
21         else p_need[ids] = need[ids] / total_need	
22       end for	
23       total_unused = be_available	
24       rem_unused = be_available	
25      [need, new_rate, rem_unused] = allocation (need, new_rate, be, p_need, total_unused, rem_unused)	
26   else	
27     new_rate[be] = allocated[be] - total_need	
28     for ids in need:	
29       if ids != be: new_rate[ids] += need[ids]	
30     end for	
31  end else	
32   else if rem_unused > 0:	
33     new_rate[be] += min(min(rem_unused, current[be]), need[be])	
34     end else	
35	
36   return new_rate	

Algorithm 3 allocUnder(allocated, current, need, unused).

1   new_rate = {}	
2   ratio = sum(need) / sum(unused)	
3	
4   for ids in need:	
5     new_rate[ids] = max(allocated[ids], current[ids])	
6     if unused[ids] > 0:	
7       new_rate[ids] = allocated[ids] - ratio * unused[ids]	
8     end if	
9   end for	
10	
11   return new_rate	

Algorithm 2

Algorithm 2 calculates the flow rate if the total needed bandwidth is more than the unused bandwidth on the link. The algorithm starts by giving out the new rate as the minimum of the allocated and the current rate (line 6) so that the current QoS is at least guaranteed to be given its allocated rate if its need for more bandwidth cannot be met immediately. On the other hand, if the current QoS is underperforming and has some bandwidth to lease out, its new rate will be set to its current rate so that the unused portion can be given to other flows exceeding its allocated rate. Every QoS class has a flag attached to it which is set to 0 (line 7) if the class does not need any excess bandwidth. If the class needs more bandwidth, the flag is set to a proportional value equivalent to the ratio of its current need and the total needed by all the QoS classes (line 8). The larger the need is for a class, the higher its needed ratio will be so that the allocation can be made fairly. The eligibility to receive more bandwidth is given to the QoS classes with higher priorities. The needed amount by these higher priority flows is given out (lines 11–13) from the unused portions of the other underperforming higher priority flows as well as from the unused portion of the best effort flow using Algorithm 4. The rate at which the best effort traffic is flowing is not touched yet.

Algorithm 4 allocation (need, new_rate, be, p_need, total_unused, rem_unused).

1  for ids in need:	
2    if ids == be:	
3      continue	
4    end if	
5    given = p_need[ids] * total_unused	
6    need[ids] -= given	
7    new_rate[ids] += given	
8    rem_unused -= given	
9  end for	
10	
11  return [need, new_rate, rem_unused]	

If there are still some high priority flows that need more bandwidth, the algorithm attempts to satisfy their requirements from the best effort traffics allocation (lines 15–33). However, if the best effort traffic needs more bandwidth, its need can be satisfied at the end (lines 34–37) if no higher priority traffic needs any more bandwidth.

For the case of the higher priority traffic needing more bandwidth after the initial allocation, the algorithm takes away the allocation for the best effort traffic (line 18) and calculates the new ratio (lines 19–22) at which the higher priority flows can be given more bandwidth. The allocation is made again using Algorithm 4 but only with the bandwidth from the best effort flow.

Algorithm 3

Algorithm 3 is used to calculate the flow rate if the total needed bandwidth is less than the unused bandwidth on the link. In this case, all the excess needs can be easily satisfied from the unused portions of the underperforming QoS classes. The algorithm first calculates the ratio of the total need and total unused for all the flows (line 2). This will be used to determine the portion which can be leased to the high performing flows.

The algorithm allocates the maximum of the current and allocated rates so that the requested bandwidth is guaranteed to be given to the QoS class. Then for the flows which are underperforming, their portions are deducted based on the ratio. This is to ensure that if there is a sudden rise in the demand from the underperforming flows, the class will have room to grow and will not incur any packet losses immediately. This leaves a margin for the traffic to grow.

Algorithm 4

Algorithm 4 allocates more bandwidth to high priority flows based on the unused and needed traffic amounts. The algorithm adjusts and returns the need and new rate matrices. The best effort traffic is not considered because any left-over traffic at the end can then be allocated to the best effort flow.

Complexity analysis

AFQoS

The four algorithms are nested as shown in Table 2.

Table 2 Time complexity of AFQOS.

Algorithm #	Algorithm name	Time complexity	
1	meterAllocation	O(n)	
2		allocUnder	O(n)	
3		allocOver	O(n)	
4			allocation	O(n)	

Algorithm 4 is the lowest level of function call and has a for loop running at line 1 in the order of n, where n is the number of QoS classes. The time complexity of this function is O(n).

Algorithm 3 also has a for loop at line 4 which runs in the order of n, giving a time complexity of O(n) as well.

Algorithm 2 has a series of non-nested O(n) executions. The first one is at line 5, and the other two at lines 12 and 25 due to the function calls to Algorithm 4. The gives a time complexity of O(n) as well as they are executing in series.

Algorithm 1 has Algorithms 2 and 3 as subprocedures where one is executed at a time. Its time complexity is carried up from Algorithms 2 and 3, giving a time complexity of O(n). The time complexity of the entire allocation algorithm is O(n). This is also the every-case time complexity.

AQoS

The AQoS algorithm employs a simple code logic at the expense of checking for unused bandwith while allocating the new rates. This gives a nested structure with each loop running in the order of O(n), giving a time complexity of (n2).

Testbed setup

The virtual network setup used in the experiment was hosted on a machine with the specifications listed in Table 3.

Table 3 Specifications of the PC used in the experimentS.

Parameter	Specification	
System type	64-bit x64 based processor	
Processor	6-core Intel(R) Core(TM) i7-9750H CPU at 2.60 GHz	
RAM	16 GB DDR4	
Storage	500 GB SSD for storage and swap	

The virtualized SDN network in Fig. 1 was created in a standalone Mininet (2022) with the specifications listed in Table 4.

Figure 1 Network topology used for testing the allocation and redistribution algorithms.

Table 4 Specifications of mininet.

Parameter	Specification	
Version	Mininet 2.3.0	
Processor	6 processors, borrowed from host machine	
RAM	4 GB	
Host	Ubuntu 20.04 LTS VM	

Ryu SDN controller was installed in Ubuntu and was connected to the Mininet as a remote controller.

The virtual network consisted of four PCs on each side of the OpenFlow switches. The PCs on switch 4 acted as servers and the PCs on switch 1 connected to the servers to query and generate flows. Each flow path consisted of a client and server PCs between which data was generated using iPerf (IPerf, 2023). All the flows generated between the client and server passed through switches 2 and 3. These switches acted as the congestion point in the network and the controller queried these switches for testing the proposed algorithms.

Methodology

We start the experiment with the objective of designing an algorithm for managing the bandwidth allocation between different QoS classes while ensuring the least packet loss per class. Thus, the experiment consisted of an OpenFlow network simulator, iPerf for generating data, a database to store the flow statistics, a Ryu controller, and OpenFlow switches with metering and flow stats capabilities.

A custom python script was written to create the topology in Fig. 1, and four flows with different rates were created between the source and destination devices. The flows were made bursty with additional logic implemented on top of the iPerf generator to simulate the bursty nature of the traffic. A meter is attached to each flow to classify them into different QoS classes and control each class’s rate.

Iperf was used to set up and generate traffic between each pair of source and destination hosts in Mininet. Four flows were set up and classified into four QoS classes, with one class for the best effort traffic. The flows were generated with different rates such that the link utilization ranges from underutilization to full utilization to overutilization. Five different traffic rates were configured for each class to test different traffic patterns.

The proposed algorithm was implemented as a controller application and run against different flows to test its effectiveness. The predictive method of rate approximation was used to approximate the flow rate of the flows for different values of capture duration. The flows were generated with different rates ranging from empty link to full saturation to exceeding capacity. For each of the rates, the experiment was run 30 times where each trial lasted for 300 s with 20 s pause for the controller to sense this pause and update the flow parameters.

For each trial of the experiment, the controller collected the flow statistics from both switch 1 and switch 2 and calculated the approximate flow rate per QoS class. The proposed algorithm then calculated the bandwidth to allocate to each QoS class and configured the switch with the new rate. The data collected from both switches were used to plot the losses for each QoS class and the overall packet loss.

Results and discussion

The same data that powered the proposed algorithm’s core logic was logged into the database for analysis. The proposed algorithm was tested against the Adaptive QoS (AQoS) algorithm in Boley, Jung & Kettimuthu (2017). Simulation results show that the proposed algorithm (AFQoS) yields a higher throughput with less packet loss per QoS class. The traffic rates for QoS 1, QoS 2, QoS 3, and QoS 4 were initialized to 8,000, 6,000, 4,000, and 2,000 kbps, respectively. The total capacity for the link was 20,000 kbps and QoS 4 was set as the best effort class.

Bandwidth allocation

The bar graphs in Figs. 2–9 show how the bandwidth allocation was done using both the AQoS and AFQoS. The allocation shows that the proposed algorithm allocates the bandwidth in an optimal way that not only incurs the least packet loss but also leaves some room for the flows to grow if there is a demand for more traffic for the current QoS class. The graphs show the instantaneous bandwidth allocation at a random time to show how the algorithm did the allocation in reaction to the current rate. With QoS in place, there could be different cases where different rate calculations may be applied. At any time during the flow session, an OpenFlow switch could either have few flows passing through, in which case, the link will be underutilized. The second case is when the link is operating at its peak capacity, in which case, the link is fully utilized. The last case is when different classes are competing for bandwidth and are incurring some packet losses. This is when the link is overutlized and there is contention for bandwidth. These three cases need different attention as to how allocations will be done.

Figure 2 Bandwidth allocation using AQoS algorithm at 7,000, 5,000, 3,000, and 1,000 kbps.

Figure 3 Bandwidth allocation using AFQoS algorithm 7,000, 5,000, 3,000, and 1,000 kbps.

Figure 4 Bandwidth allocation using AQoS algorithm at 8,000, 6,000, 4,000, and 2,000 kbps.

Figure 5 Bandwidth allocation using AFQoS algorithm at 8,000, 6,000, 4,000, and 2,000 kbps.

Figure 6 Bandwidth allocation using AQoS algorithm at 9,000, 7,000, 5,000, and 3,000 kbps.

Figure 7 Bandwidth allocation using AFQoS algorithm at 9,000, 7,000, 5,000, and 3,000 kbps.

Figure 8 Bandwidth allocation using AQoS algorithm at 7,000, 5,000, 3,000, and 4,000 kbps.

Figure 9 Bandwidth allocation using AFQoS algorithm at 7,000, 5,000, 3,000, and 4,000 kbps.

Case 1: each QoS class, as well as the total link, is underutilized

Figure 2 shows how the bandwidth was allocated using the AQoS algorithm at five consecutive time intervals. At time t = 0, the current requirement for QoS 1, QoS 2, QoS 3, and QoS 4 were 6,086, 4,594, 3,188, and 904 kbps with the traffic set to flow at an average rate of 7,000, 5,000, 3,000, and 1,000 kbps respectively. In this case, the requirements were below the maximum allowed bandwidth thus, the requirements were satisfied. However, due to the nature of the algorithm, it seizes the unused bandwidth, which results in the QoS classes rate being capped to their current requirement. A similar allocation happens for the time intervals t = 1, 2, 3, and 4.

Figure 3, on the other hand, shows how the bandwidth was allocated using the proposed algorithm. At time t = 0, the current requirement for QoS 1, QoS 2, QoS 3, and QoS 4 were 6,633, 4,726, 2,836, and 1,057 kbps, respectively. All the QoS class rates were within the initial configured rates in this case. Thus, there is no change in the rates. It is left to its initial allocation. This also ensures that each class has enough spare bandwidth to grow in case there is a sudden increase in the current requirement. The same allocation rules are used for the time intervals t = 1, 2, 3, and 4.

Case 2: each QoS class, as well as the total link, is approximately at full utilization

Figure 4 shows how the bandwidth was allocated using the AQoS algorithm at five consecutive time intervals. At time t = 0, the current requirement for QoS 1, QoS 2, QoS 3, and QoS 4 were 8,253, 5,400, 3,861, and 1,779 kbps with the traffic set to flow at an average rate of 8,000, 6,000, 4,000, and 2,000 kbps respectively. In this case, the requirements were, on average, at their full capacity with the link at maximum utilization. Some classes were generally above their initial allocation, while some were below their initial allocation. With the AQoS algorithm, the underperforming classes were leased out to other classes that needed more bandwidth. At time t = 0, since QoS 1 required more bandwidth, it was easily catered for by QoS 2, QoS 3, and QoS 4 as it was underperforming. However, there is no room for QoS 2, QoS 3, and QoS 4 to grow if they need more bandwidth. The link is also not fully utilized, as those below their initial allocation are capped at their current rate. A similar allocation happens for the time intervals t = 1, 2, 3, and 4.

Figure 5, on the other hand, shows how the bandwidth was allocated using the proposed algorithm. At time t = 0, the current requirement for QoS 1, QoS 2, QoS 3, and QoS 4 were 8,186, 5,136, 4,634, and 1,982 kbps, respectively. In this case, since some bandwidth was available in the link, the extra requirement from QoS 1 and QoS 3 were safely satisfied by QoS 2 and QoS 4. However, since the initial allocation of QoS 2 and QoS 4 were 6,000 and 4,000 kbps, respectively, the unused bandwidth was not given out to those in need. Some extra margin was left out in QoS 2 and QoS 4 for them to grow should there be an increase in their current requirement. The same allocation rules are used for the time intervals t = 1, 2, 3, and 4.

Case 3: all QoS classes are exceeding their allocation and the link is bottlenecked

Figure 6 shows how the bandwidth was allocated using the AQoS algorithm at five consecutive time intervals. At time t = 0, the current requirement for QoS 1, QoS 2, QoS 3, and QoS 4 were 8,515, 8,034, 5,225, and 2,548 kbps with the traffic set to flow at an average rate of 9,000, 7,000, 5,000, and 3,000 kbps respectively. In this case, all the QoS classes were exceeding their initial rate. The AQoS algorithm captures the bandwidth from QoS 4 (best effort class) and leases it out to the high priority traffic. QoS 1 is first catered for with the remaining bandwidth being passed down to QoS 2. QoS 2 still however needs more bandwidth which cannot be satisfied right away as the link is full. QoS 3 also needs more bandwidth however, its rate is set to its initial rate as the link is full. The packet losses for QoS 1 will be the lowest as its requirement has been satisfied. However, the other classes will incur significant packet losses as their requirement was not met. A similar allocation happens for the time intervals t = 1, 2, 3, and 4.

Figure 7, on the other hand, shows how the bandwidth was allocated using the proposed algorithm. At time t = 0, the current requirement for QoS 1, QoS 2, QoS 3, and QoS 4 were 9,982, 7,773, 5,380, and 2,538 kbps, respectively. In this case, all the classes are again exceeding their initial allocation. The algorithm leases out the bandwidth from QoS 4 class however it does not fully allocate the bandwidth to QoS 1. It gives out a little bandwidth to all the priority traffic based on the priority while ensuring that the allocated amount yields the least packet loss for each class. The same allocation rules are used for the time intervals t = 1, 2, 3, and 4.

Case 4: priority QoS classes are within their initial allocation, but best effort flows require more bandwidth

Figure 8 shows how the bandwidth was allocated using the AQoS algorithm at five consecutive time intervals. At time t = 0, the current requirement for QoS 1, QoS 2, QoS 3, and QoS 4 were 6,719, 5,561, 2,529, and 4,630 kbps with the traffic set to flow at an average rate of 7,000, 5,000, 3,000, and 4,000 kbps respectively. In this case, all the QoS classes were underperforming. However, QoS 4 (best effort flow) requires more bandwidth. Since the priority flows have unused bandwidth, the AQoS algorithm caps the priority flows rate to its current rate and leases out the unused bandwidth to the best effort class. The priority classes do not have room to grow if more bandwidth is needed. Thus, it is expected to cause more packet loss when traffic demand increases before a new allocation can be made. A similar allocation happens for the time intervals t = 1, 2, 3, and 4.

Figure 9, on the other hand, shows how the bandwidth was allocated using the proposed algorithm. At time t = 0, the current requirement for QoS 1, QoS 2, QoS 3, and QoS 4 were 7,809, 4,109, 3,497, and 3,356 kbps, respectively. In this case, all the priority classes are underperforming. However, the best effort class needs more bandwidth. The algorithm takes some unused bandwidth from the priority flows and leases it to the best effort class. However, it ensures that the priority flows are not constrained to their current rate and have some unused bandwidth to grow into if there is a demand for more bandwidth from the priority flows. The same allocation rules are used for the time intervals t = 1, 2, 3, and 4.

Throughput/packet loss for AQoS and AFQoS

Figures 10–13 shows the throughput achieved by both algorithms for different allocation and traffic demands. The AFQoS algorithm always leaves some unused bandwidth for the current QoS class to grow into if there is a need for more bandwidth. The AQoS algorithm, however, caps the bandwidth to the class’s current rate if it has some bandwidth to lease out. These major differences in both algorithms cause a significant packet loss difference throughout the flow. The graphs show the throughput for four different traffic conditions. The total link capacity was 20,000 kbps.

Figure 10 Throughput gained for both algorithms at 7,000, 5,000, 3,000, and 1,000 kbps.

Figure 11 Throughput gained for both algorithms at 8,000, 6,000, 4,000, and 2,000 kbps.

Figure 12 Throughput gained for both algorithms at 9,000, 7,000, 5,000, and 3,000 kbps.

Figure 13 Throughput gained for both algorithms at 7,000, 5,000, 3,000, and 4,000 kbps.

Case 1: each QoS class, as well as the total link, is underutilized

Figure 10 shows the throughput gained by both AQoS and AFQoS algorithms with QoS 1, QoS 2, QoS 3, and QoS 4 traffic flowing at an average rate of 7,000, 5,000, 3,000, and 1,000 kbps, respectively. The total bandwidth requirement was 16,000 kbps, with the link at 80% utilization on average. The throughput gained by AQoS algorithm for QoS 1, QoS 2, QoS 3, and QoS 4 (BE) was 90.22%, 90.2%, 89.8%, and 88.42%, with an average throughput of 89.66%. The throughput gained by AFQoS algorithm for QoS 1, QoS 2, QoS 3, and QoS 4 (BE) was 99.99%, 99.99%, 99.99%, and 99.95%, with an average throughput of 99.98%, on average. The results show that the AQoS algorithm did not run at full capacity even when the link was underutilised. This is because the algorithm caps all the classes at their current rate, leaving no room for the flows to grow as demand increases. The proposed algorithm, however, achieved a near 100% throughput on an underutilized link.

Case 2: each QoS class, as well as the total link, is approximately at full utilization

Figure 11 shows the throughput gained by both AQoS and AFQoS algorithms with QoS 1, QoS 2, QoS 3, and QoS 4 traffic flowing at an average rate of 8,000, 6,000, 4,000, and 2,000 kbps, respectively. The total bandwidth requirement was 20,000 kbps, with the link at 100% utilization on average. The throughput gained by AQoS algorithm for QoS 1, QoS 2, QoS 3, and QoS 4 (BE) was 90.73%, 90.5%, 90.33%, and 81.5%, with an average throughput of 88.23%. The throughput gained by AFQoS algorithm for QoS 1, QoS 2, QoS 3, and QoS 4 (BE) was 99.99%, 99.96%, 99.99%, and 92.02%, with an average throughput of 97.99%, on average. The results show that at full link capacity, the AQoS algorithm performs similarly to an underutilized link. The AFQoS algorithm, however, performs well on a fully utilized link, with very little packet loss and achieving a near 100% throughput.

Case 3: all QoS classes are exceeding their allocation and the link is bottlenecked

Figure 12 shows the throughput gained by both AQoS and AFQoS algorithms with QoS 1, QoS 2, QoS 3, and QoS 4 traffic flowing at an average rate of 9,000, 7,000, 5,000, and 3,000 kbps, respectively. The total bandwidth requirement was 24,000 kbps, with the link at 120% utilization on average. The throughput gained by AQoS algorithm for QoS 1, QoS 2, QoS 3, and QoS 4 (BE) was 90.68%, 90.62%, 90.24%, and 40.88%, with an average throughput of 78.11%. The throughput gained by AFQoS algorithm for QoS 1, QoS 2, QoS 3, and QoS 4 (BE) was 99.83%, 99.11%, 99.53%, and 48.93%, with an average throughput of 86.85%, on average. The results show that the AQoS algorithm performs similar to previous cases on an overutilized link. The priority QoS classes cap at similar throughput; however, the best effort classes throughput has decreased as a result of bandwidth leasing. The AFQoS algorithm performs well for priority traffic and achieves nearly 100% throughput. The best effort class however, achieves a lower throughput due to bandwidth leasing and a higher overall throughput for the link.

Case 4: priority QoS classes are within their initial allocation, but best effort flows require more bandwidth

Figure 13 shows the throughput gained by both AQoS and AFQoS algorithms with QoS 1, QoS 2, QoS 3, and QoS 4 traffic flowing at an average rate of 7,000, 5,000, 3,000, and 4,000 kbps, respectively. The total bandwidth requirement was 19,000 kbps, with the link at 95% utilization on average. The throughput gained by AQoS algorithm for QoS 1, QoS 2, QoS 3, and QoS 4 (BE) was 90.32%, 90.31%, 89.97%, and 88.21%, with an average throughput of 89.70%. The throughput gained by AFQoS algorithm for QoS 1, QoS 2, QoS 3, and QoS 4 (BE) was 99.99%, 99.99%, 99.99%, and 93.7%, with an average throughput of 98.42%, on average. The results show that on an underutilized link with the best effort flow in need of more bandwidth, the AQoS algorithm is able to cater for the best effort flow and provide a similar throughput as the priority classes. The AFQoS algorithm performs well for priority classes and achieves a good throughput for the best effort class as well.

Conclusion

This article proposed an algorithm for solving the bandwidth allocation problem using the flow statistics from the OpenFlow switches. The algorithm queries the switch at certain intervals to get the current flow details to calculate the proportion of bandwidth to allocate to each QoS class. In literature, various approaches have been proposed. However, this article focuses on a specific subset of a problem. The algorithm is used on specific switches that meet certain network conditions.

The proposed algorithm was compared with the AQoS algorithm for performance and efficiency. The algorithm was tested against different traffic scenarios, such as when the link is underutilized, at full capacity, and overutilized. It was noted that the proposed algorithm not only achieves a higher overall throughput but also causes the least packet loss per QoS class.

The algorithm was developed as a controller application and can easily be integrated into any OpenFlow network requiring QoS provisioning. Although the algorithm achieves a near maximum throughput for various traffic scenarios, it was noted that when the best effort class needs more bandwidth, the algorithm cannot optimally allocate the bandwidth despite the link not running at full capacity. Thus, in future work, the authors intend to rectify and improve the allocation efficiency of the algorithm. The research was conducted in Mininet simulation to mimic the real network behavior however, it has not been tested on a real network with SDN supported routers. In future work, the authors also intend to test the effectiveness of the algorithm on real devices.

Additional Information and Declarations

Competing Interests

Author Contributions

Data Availability

The authors declare that they have no competing interests.

Krishneel Deo conceived and designed the experiments, performed the experiments, analyzed the data, performed the computation work, prepared figures and/or tables, authored or reviewed drafts of the article, and approved the final draft.

Kaylash Chaudhary conceived and designed the experiments, performed the experiments, analyzed the data, performed the computation work, prepared figures and/or tables, authored or reviewed drafts of the article, and approved the final draft.

Mansour Assaf conceived and designed the experiments, performed the experiments, analyzed the data, performed the computation work, prepared figures and/or tables, authored or reviewed drafts of the article, and approved the final draft.

The following information was supplied regarding data availability:

The code and data generated during the experiment are available at Zenodo:

https://zenodo.org/records/10184077.

-Deo, K. (2023). Adaptive Quality of Service using Predictive Rate Approximation using OpenFlow Meters. In PeerJ Computer Science. Zenodo. https://doi.org/10.5281/zenodo.10184077.

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
