# Peer review of "Adaptive quality of service for packet loss reduction using OpenFlow meters"

_PeerJ Computer Science, doi:10.7717/peerj-cs.1848_

## Round 0.1 · original submission · Major Revisions

Dear Dr. Deo,

Thank you for your submission to PeerJ Computer Science.

Based on the reviews of qualified reviewers, which you will find below or attached.

I recommend your paper be revised according to the reviewers' comments and resubmitted.


-Best regards,
Ahyoung.

Reviewer 1 ·

Basic reporting

The manuscript deals with a simple algorithm to share available capacity among a set of competing flows in a guaranteed service SDN. The paper has a clear, logical structure, and is well written, but the quality of the figures is low and should be improved with better resolution. Also, though the specification of the algorithms is clear, the code should be made available in order to replicate the results by other researchers.

While the problem is clearly stated and described, the motivation is rather weak: first, there is a large body of literature about fair queuing algorithms, all of which constitute approximations to the GPS (Generalized Processor Sharing) discipline that the authors intend to mimick for using the excess capacity in a fair manner. These fair queueing disciplines can be directly implemented in the SDN framework. Secondly, while the authors claim several times that their method is optimal and minimized the packet losses, this property is not substantiated or supported at all, neither formally nor experimentally,

The bibliographic references are correct and up to date.

Experimental design

In simple terms, the proposal in this work is a weighted share of the excess capacity in a single link among competing flows, where the weighs are related to the priorities if the flows (linearly).

1) The mechanism can be easily described with a couple of simple formulas, so Algorithms 1-4 are a little over specified, in my opinion.

2) The claim of optimality of the algorithm, which is mentioned several times at the beginning and at the end of the manuscript, is not demonstrated, theoretically or practically. The same applies to the minimization of the packet losses, the proposed method does not guarantee in advance that losses are minimized.

3) Though a linear share of the excess bandwidth might seem reasonable in most cases, the authors do not discuss at all the issue of fairness in bandwidth allocation, which is directly related ti their proposal.

4) Methodologically, the experimental part is a succession of 4 test cases (underloaded, critically loaded, and overloaded) for very specific configurations. It is difficult, in my opinion, to draw general conclusions only from these limited test scenarios.

5) The presentation of the results is poor. It is advised to improve the quality/resolution of the plots and graphics, and to make experiments in a more complex network setting.

As a result of the above points, the conclusions are only weakly supported by the results.

Validity of the findings

As mentioned previously, the results do not have a strong experimental support, and can possibly be achieved with a properly chosen fair queueing discipline, which already account for the fair distribution of the bandwidth in excess in a given time. There is also a point for further conceptual discussion: the SDN principle is well suited to the problems of medium-scale traffic engineering (path selection, QoS provision, etc.), but the intent in this work is to leverage on SDN statistics and programmability to perform a network reconfiguration at a small timescale (at each second, say), which is slightly at odds with the SDN philosophy. In other words, using SDN rules to implement a queueing policy with bandwidth guarantees is a kind of overuse, at some extent.

As said before, the source code should be made available in order to check and reproduce the results in the experiments, if possible.

Cite this review as

Reviewer 2 ·

Basic reporting

This paper (peerj-CS-87038) aims to propose the bandwidth allocation and distribution algorithm by using the flow statistics from the OpenFlow switches to allocate bandwidth to different QoS classes. It is interesting and some simulation results are provided. Here are some comments:
1. The motivation is not clearly given in the introduction part. The references are mostly dated and the contributions over the more recent works are unclear, and research developments about the related topics are not fully reviewed.
2. The presentation of the paper needs to be much improved. There are some mistakes such as "Figure 0.1- Figure 0.12, Figure 7.7, or Figure 7.9-7.12". Figure title or number should be improved and unified. The paper should be aligned on both ends. Reference citation should not appear in the Abstract.
3. The format of some references should be unified and meet requirements. Some sources of literature are missing. Moreover, in some references, the first letter of the title of the paper is only capital. In others, all first letters of the titles of the paper are capital. Please maintain consistency.

Experimental design

no comment

Validity of the findings

4. Theoretical contribution should be enhanced. Complexity analysis of the proposed algorithm (AFQoS) should be given in the text compared to the existing results such as AQoS.

Additional comments

5. Authors should have provided a better discussion of the cited references. Many references such as [13-19] have been reported, but without providing a critical view of them.
6. The relevance to Peer J Computer Science journal should be enhanced. Over the 26 references, there is no article related to PeerJ Comput. Sci..

Cite this review as

Reviewer 3 ·

Basic reporting

No comment

Experimental design

No comment

Validity of the findings

No comment

Additional comments

The authors in this paper propose a bandwidth allocation and distribution algorithm that employs the flow statistics from the OpenFlow switches to allocate bandwidth to different QoS classes on optimal basis. The proposed algorithm uses the SDN controller's flow monitoring component to query the flow statistics from the switch to first approximate the traffic flow rate and finally computing the optimal bandwidth values for assigning a particular QoS class. The algorithm may be applied to OpenFlow compliant switches in the path of communication. The results findings suggest that the proposed algorithm achieves an average of 9% performance gain compared to the AQoS algorithm. In my opinion, the proposed scheme seems to be viable with convincing findings. However there are few minor concerns that must be addressed before any recommendation.
1. The citation should be eliminated from the Abstract.
2. The scheme organization in section I seem to have a missing section 8 with conclusion.
3. It is advisable for the authors to summarize the literature work in the last paragraph of the same section.
4. In section 2, the authors should provide a table illustrating a summary of the discussed schemes with demerits.
5. Please rewrite the line 215, “The packets dropped by each classes rate limiter is minimum.” with improved grammar.
6. If the format of the journal permits, the manuscript could better be right aligned.
7. Please cite the virtual network “Mininet” as well as “iperf” appropriately in section 5.
8. It is advisable to specify the testbed parameters in tabular form for more clarity.

Cite this review as

---

## Round 0.2 · Minor Revisions

Based on the reviews of qualified reviewers, which you will find below or attached. please clearly address these changes and resubmit within the next 10 days.

Reviewer 2 ·

Basic reporting

no comment

Experimental design

no comment

Validity of the findings

no comment

Additional comments

This revised paper (peerj-CS-87038-v1) has been improved. Most concerns have been addressed. The reviewer can recommend its acceptance for publication with minor revision. But the authors should carefully check the presentation. For example, the title of Section 2"Literature" can be improved. "10. Reference"(page 23, line 634) should be revised as References.

Cite this review as

Reviewer 3 ·

Basic reporting

no comment

Experimental design

no comment

Validity of the findings

no comment

Additional comments

The editorial quality has been improved and I believe the paper's contribution warrants acceptance.

Cite this review as

---

## Round 0.3 · accepted · Accept

I confirm that the authors have addressed all of the reviewer's comments.